# Study on the Incorporation of Oat and Yeast β-Glucan into Shortbread Biscuits as a Basis for Designing Healthier and High Quality Food Products

**DOI:** 10.3390/molecules27041393

**Published:** 2022-02-18

**Authors:** Anna Zbikowska, Malgorzata Kowalska, Katarzyna Zbikowska, Sylwia Onacik-Gür, Urszula Łempicka, Paweł Turek

**Affiliations:** 1Faculty of Food Assessment and Technology, Institute of Food Sciences, Warsaw University of Life Sciences (WULS-SGGW), Nowoursynowska St. 159c, 02-776 Warsaw, Poland; urszula_lempicka@gmail.pl; 2Institute of Food Sciences, Faculty of Chemical Engineering and Commodity Science, Kazimierz Pulaski University of Technology and Humanities, Chrobrego St. 27, 26-600 Radom, Poland; mkowalska7@vp.pl; 3Faculty of Medicine, Medical University of Warsaw, Żwirki i Wigury St. 61, 02-091 Warsaw, Poland; 4Department of Meat and Fat Technology, Prof. Waclaw Dabrowski Institute of Agriculture and Food Biotechnology—State Research Institute, Rakowiecka 36 St., 02-532 Warsaw, Poland; sylwia.onacik@gmail.com; 5Department of Non-Food Product Quality and Safety, Cracow University of Economics, Rakowicka St. 27, 31-510 Cracow, Poland; turekp@uek.krakow.pl

**Keywords:** saturated fatty acids, dietary fiber, quality of shortbread biscuits

## Abstract

According to international health and food organizations and authorities, people should limit fat intake since fat is the most caloric component of food and it is often a source of unsafe saturated fatty acids (FA) and trans isomers. The greatest health benefits come from replacing shorts with dietary fiber molecules. The aim of the study was to determine the possibility of reducing shortening content, which has an undesirable profile of FA, by addition of β-glucan molecules in shortbread biscuits. The effect of oat and yeast β-glucan supplementation on physical and sensory quality of products with reduced fat content (max 15%) were studied. It was shown that the substitution of shortening by β-glucan in shortbread biscuits is possible to a limited extent. Reduction in product energy value (up to 36 kcal/100 g) and content of undesirable FA (maximum 2.1 g/100 g) and increased of β-glucan content, regardless of the type, caused deterioration of biscuits quality and affected changes during storage. The substitution of shortening by β-glucan in food is a good way to improve nutritional value by increasing the amount of dietary fiber molecules, reducing calories, and amount of SFA in diets.

## 1. Introduction

Lipids belong to the group of macronutrients provided by a diet. They mainly include triacylglycerol molecules, but also phospholipids, glycolipids, and cholesterol [1,2]. These molecules are an essential component of a diet and processed foods. Lipids provide energy, essential fatty acids, and vitamins (A, D, E, K). In addition, they improve the taste and texture of food and make it easier to swallow [2]. These compounds fulfill many roles in living organism. They are structural components in organism, energy storage to fuel metabolism, participate in signal transduction [3].

However, lipids, which are a component to food such as shortbread cookies are considered unhealthy because they are the most energetic nutrients in food and are a source of saturated fatty acids (SFA) (usually 40%) and sometimes also trans fatty isomers (TFA) [4,5,6]. High consumption of SFA and TFA is associated with an increased serum Low-Density Lipoprotein, which consequently can lead to coronary heart diseases (CHD) [7]. Cardiovascular diseases can be countered by a drastic reduction in the consumption of foods rich in saturated fats acids [8,9]. For this reason, much attention is focused on improvement of nutritional profiles of these products by development of low-fat products using fat replacers [10]. Moreover, substitution of fats rich in SFA or TFA in food may contribute to the prevention of overweight and development of obesity [11]. According to the World Health Organization, around 40% of adults are overweight [12]. In addition, due to the negative impact of TFA on the body, the European Commission Regulation 2019/649 was in force since 2021, limiting the content of industrial TFA in fat (up to 2 g TFA/100 g of product) [13]. Therefore, it is advisable to replace industrial shortenings in processed food with ingredients with high biological potential, e.g., dietary fiber biomolecules [14,15,16,17,18]. It was reported that many fibers with a nutritional value could be used to substitute fat in bakery products, such as inulin, fruits, and vegetable fibers, cereal fiber, and β-glucans [19,20]. According to Onacik–Gur et al. [20], substitution of fat up to 20% with barley β-glucan in muffins was possible; however, addition of it causes a quality deterioration [20]. 

Due to its pro-health properties, β-glucan deserves special attention. Depending on the origin, they differ in terms of biological activity and chemical structure. β-glucans are organic chemicals compounds from the group of polysaccharides, derived from cereals. They lower cholesterol and blood sugar and contribute to the maintenance of healthy body weight [21]. The documented effect of this nutrients on blood cholesterol concentration allowed the use of health claims on foods containing at least 1 g of glucan derived from oats or barley per given portion of the product [22]. In turn, glucans isolated from yeast have anti-infective, anticarcinogenic, and immunostimulatory effects. β-glucan, regardless of its origin, has prebiotic properties—it supports the development of beneficial intestinal microflora [23,24,25]. 

Considering the purposefulness of eliminating fats that increase the risk of cardiovascular diseases from the diet and introducing nutrients with increased biological potential in their place, this seems to be a desirable solution. The study attempts to replace commercial shortening with β-glucans biomolecules from two different sources in confectionery products. Each fat replacer may influence the sensory and physical quality of bakery goods different. It is important for food technologists developing new bakery products with fat replacers to understand the effects of yeast and oat β-glucan on quality. The impact of such reformulation on the quality of dough and final product was analyzed, and, in particular, it was analyzed how the replacement of shortening with β-glucans affects kinetics of texture changes, kinetics of color degradation, total color difference (ΔE), chroma, and browning index (BI) during biscuits storage with a different proportion of β-glucans.

## 2. Results and Discussion

### 2.1. Characteristic of Raw Lipids

When analyzing the nutritional value of the lipid used in the study, the share of desirable fatty acids (DFA) (sum of unsaturated fatty acids (USFA)) and undesirable fatty acids (OFA) (sum of saturated fatty acids (SFA) and trans fatty acids (TFA)) was taken into account. The highest amount of SFA was in the shortening, and therefore the ratio of DFA to OFA did not exceed 1 (Table 1), which is typical for fats solid at room temperature (20–25 °C). The solid consistency of fats rich in saturated fatty acids and high content of solid phase make it a very good raw material in the production of shortbread biscuits. Shortenings should have appropriate technological properties, good plasticity, solid consistency, and presence of solid phase. These parameters are positively influenced by the presence of SFA in the lipid composition, which is characterized by a high melting point [26,27]. In the analyzed shortening, palmitic acid dominated in the SFA group, followed by stearic acid. These are typical components of the lipid fraction of confectionery products [4]. Long-chain SFA may contribute to an increased risk of cardiovascular disease [28] and an increase in body weight [29]; therefore, it should be eliminated from food. The commercial shortening used in the study was characterized by a low content of free FAs and a low degree of oxidation (Table 1), and therefore it was also suitable for processing at high temperatures. The analyzed lipid was characterized by a relatively high values of the atherosclerotic index—above 1 (Table 1), which is typical for shortenings and proves that they are products with low nutritional value [10].

### 2.2. Nutritional Value of Lipids in Shortbread Biscuits and Their Density and Geometry

Reducing energetic value of the product, which traditionally contains large amounts of fat, is important in view of overweight and obesity, common in societies of developed countries [30,31]. The introduction of fiber in place of fat reduces the calorific value of processed food and increases its nutritional value. β-glucan is almost unutilized in human digestion tract and functions as a non-caloric food ingredient [32]. Introducing β-glucan to shortbread biscuits resulted in lowering the lipid content in the product by 5% (samples A1 and B1), 10% (A2 and B2), and 15% (A3 and B3), compared to that of the control sample (0). Fat reduction by 5% to 15% resulted a significant decrease in the energy value of the products (from 11.7 kcal to 36 kcal). It is important due to the high popularity of this kind of snacks [3] and problems with overweight and obesity (according to the WHO in 2021). Moreover, the content of undesirable fatty acids decreased by 2.1 g per 100 g of the product (Section 3.1), which led to obtaining a healthier product. In accordance with the recommendations of the Committee for Nutrition, the quantity of fat in the diet should be limited, and TFA should be eliminated [33].

However, from the point of view of food production, fat has a huge impact on production process of shortbread pastry and final quality of products. Insufficient fat prevents shortbread dough preparation (<15%) without adding a component, making it possible to obtain good quality products [14]. Reducing the fat content in shortbread biscuits and replacing it with β-glucan preparations influenced the geometry, density, and spread ratio (SI) of biscuits (Table 2). It can be assumed that the obtained results are related to the properties of the fibers. The absorption of the water present in the dough by β-glucan leads to a lower air cells expansion and an increase in density, which in turn results in a coherent consistency and makes it difficult to increase the width of biscuits during baking. Quality deterioration of the dough and baked products can also be explained by the decreasing fat content, which prevents water absorption by dough ingredients (flour, fiber) and inhibits the development of the gluten network, preventing the gluten protein molecules from contacting each other. The most similar to the control sample were products in which fat was reduced by 5%, and in its place, oat β-glucan was introduced. Worse effects were achieved with the use of microbial β-glucan. The reason for higher density of products with microbial rather than oat β-glucan is the former’s poor ability to dilute in water. Particles of microbial β-glucan weigh biscuits down, and they prevented gas cells to expand during baking. Also, other studies showed that the elimination of lipids from confectionery products is associated with a deterioration in quality [34]. Other authors also reported a negative impact (decrease in volume and height) of fat substitution by β-glucans of various origins on the physical quality of fatty sponge cake products [16,21,35]. This is because fat molecules are an important component of baked goods: they hold the air in raw dough, thereby increasing its volume and creating a structure in the final product [19,36].

### 2.3. Color Parameters Assessment 

Particularly important task during the development of shortbread biscuits is to maintain the characteristics of these products that are significant for consumers, especially over a long period of storage time. Color parameters of the cookies changed statistically significantly depending on the amount of β-glucan addition. Parameters L* (white/black) and b* (yellow/blue) decreased for cookies with increasing content of β-glucan powder. Microbial β-glucan had a greater effect on color (samples B1–B3). The brightest were in the control sample, which were significantly different (pv < 0.05) than the others (Table 3). Darker color is related to the nonenzymatic browning forming during baking: the Maillard reaction and sugar caramelization. A higher value of the L* parameter (brightness) indicates that a lower fat content and a higher proportion of added polymer lead to a reduction in these reactions. The color of the shortbread biscuits may also be affected by the color of the preparation itself. The color of the yeast β-glucan preparation was much darker (brown) than that of the oat β-glucan preparation (cream). Moreover, nonhydrolyzed insoluble β-glucan molecules do not participate in the Maillard reaction as it is a long-chain polysaccharide.

In the case of a* parameter (red/green), the applied additive increased its value. It shows that baked shortbread biscuits became increasingly green–blue with increasing content of β-glucan in the recipe. Similar results were obtained by Gouveia et al. [4], who added microalgae to cookies. Similarly, Abozeid et al. [37] observed an increase in the share of red color in cookies, where the fat was replaced with the addition of pectin and egg white.

Total difference in color (ΔE) and browning index (BI) value increased in baked products by adding of β-glucan. The largest ΔE was found in shortbread biscuits with the largest addition of microbial β-glucan (B3). If ΔE > 3, the difference in color can be observed by the human eye [38]. Therefore, all products (B1–B3) were visibly different from the control sample. 

BI indicates the purity of brown color, which is particularly important in the case of baked cookies [39], where enzymatic and nonenzymatic browning takes place. This index increased with the addition of β-glucans (Table 3). Chroma, i.e., intensity of color [40], indicates the saturation, and the highest values were obtained for the control sample (without β-glucan addition) despite a strong green color of samples with β-glucan addition. Products with the lowest addition had the largest color saturation value.

Changes occurring during storage may be seen as visible differences in the color of shortbread biscuits. It was found that after the storage period, the L*, a* and b* parameters changed (Figure 1). Considering the L* parameter, a slight increase for all samples was observed after storage suggesting a brightening of the color. The decrease in fat content and the increase in the addition of β-glucan, regardless of the origin, increased the value of the L* parameter.

The a* values increased during the storage of biscuits, regardless of the variant, with the exception of control biscuits, for which the increase in this parameter was observed only between one and eight days of storage. During storage, the intensity of yellow color (b*) increased in all variants. The highest color change rate was recorded in the control shortbread biscuits.

Total color difference (ΔE*) parameter can indicate the significance of color difference visible by human eye between stored and control sample, when ΔE > 3 [41]. None of the analyzed samples showed visible changes after the storage period. The ΔE* ranged from 0.4 to 2.8 (Figure 2c); thus, based on the obtained results, the degradation processes of color were not significant in stored shortbread biscuits. Chrome and BI value increased for all variants during storage. The greatest differences were observed for products with microbial β-glucan (Figure 2a,b).

### 2.4. Texture Parameters Assessment 

On the one hand, lipids found in pastry products are a source of energy and unhealthy fatty acids [4,5,6]; on the other, they are one of the main ingredients that have a big influence on cookies texture. As discovered in earlier works, decreasing fat content or substituting fat with different components has a huge influence on texture parameters of bakery products [14,15,34,42,43]. 

The addition of β-glucan preparations changed hardness of shortbread biscuits. Oat β-glucan increased the hardness of biscuits, in contrast to microbial β-glucan (Figure 3). The differences in the influence of the β-glucans used on the texture of the cookies were caused by their origin. The increase in hardness of products with the addition of oat β-glucan may be related to the ability of this additive to absorb water, which may lead to a reduction in air cells in the dough and formation of a dense structure of products. On the other hand, the decrease in the hardness of shortbread biscuits containing a polymer of microbial origin may be related to the low solubility of this β-glucan in water and its lower water-binding capacity compared to that of the oat β-glucan. Replacing fat with various fibers most often causes an increase in the hardness of baked products [14,15,34,44]. Similarly, Kalinga and Mishra [35] showed that the addition of oat and barley β-glucan concentrates (in place of fat) increases the hardness of sponge-fat products. In their opinion, the increase in hardness was caused by the thermal coagulation of proteins during baking and reduction in dough aeration due to the loss of fat, which led to a denser crumb structure. In turn, Shyu and Sung [45] observed a decrease in hardness with increasing addition of γ-polyglutamic acid obtained from *Bacillus* spp.

During the three weeks of storage of shortbread biscuits, an increase in the hardness was found regardless of the variant (Figure 3). Faster rate of texture change was observed in biscuits with microbial polymer. Regardless of the storage time, shortbread biscuits with the highest fat substitution by oat β-glucan (A3) were the hardest, while the lowest breaking force values were recorded for the B3 samples (maximum addition of yeast β-glucan). Similarly, Kalinga and Mishra [35] observed an increase in hardness of low-fat cakes with oat and barley β-glucan during storage. They noticed a stronger increase in hardness with the introduction of barley β-glucan.

### 2.5. Analysis of Water Content in Stored Shortbread Biscuits

Most of the shortbread biscuits with increased nutritional value had a statistically significantly higher water content after baking than the control sample. Moisture increased with storage time, regardless of the variant. Greater changes in water content were observed in the case of shortbread biscuits with the addition of microbial polymer (Figure 4). Also, in other studies, the influence of other types of fat replacers (maltodextrin, polydextrose, apple fiber, inulin) on the water content in pastry products was reported [15,34,46].

The increase in water content could be caused by absorption of moisture from the environment because of package barrier properties that are too low. According to Mohebbi et al. [47] by increasing level of β-glucan addition (from 0.8% to 1.2%), there was a continuous increase in water content of bread on the first day of storage.

### 2.6. Sensory Properties of Shortbread Biscuits

Increasing the nutritional value of shortbread biscuits caused changes in their sensory quality (Table 4). The elimination of fat by the addition of the bioactive ingredient worsened the sensory quality of shortbread biscuits (Figure 5), which is related to both the reduction in fat in the composition and the introduction of β-glucan in its place. Fat is the foremost ingredient responsible for mouthfeel, lubricity, tenderness, cookie color, taste, general product appearance, and shelf life [48]. In the case of appearance and texture, less differentiation of the products was shown. Reformulation was found to have a greater impact on taste, odor, and overall quality. Introduction of yeast β-glucan caused a greater reduction in sensory quality of biscuits than in the case of shortbread biscuits with plant-based biopolymer. The smallest decrease in desirability was demonstrated with the smallest substitution of fat with the bioactive component. Also, in other works, a decrease in sensory quality of products as a result of reducing fat content by replacing it with various dietary fibers was reported [14,15,16,38,49]. According to Laguna et al. [49], substitution of more than 15% fat with inulin causes a sensory quality deterioration of baked products, especially their texture. Żbikowska et al. [14] noted replacement of 25% fat with microcrystalline cellulose generated good effects; these products were high in sensory quality. In turn, according Mohebbi et al. [47] the sensory evaluation indicated that the use of β-glucan and resistant starch (0.8%–1.2%) has no adverse effects on bread quality.

### 2.7. Correlation Analysis between the Properties of Shortbread Biscuits and the Type of β-Glucan

Based on the obtained results, correlation between content of β-glucans and values of the tested parameters of shortbread biscuits was analyzed (Table 5). Strong correlations were found between content of β-glucan, regardless of the type, and volume and width. Strong negative correlations were found between content of yeast β-glucan and color parameters (L* and b*) and the overall quality of biscuits. Inverse relationships were found in the case of water content. Strong positive correlation was found between amount of oat β-glucan and water content of biscuits than in case of biscuits with microbial β-glucan.

## 3. Materials and Methods

### 3.1. Material

Properties of dough and biscuits with increased healthiness (Table 6) were studied. The ingredients used for cookies were as follows: wheat flour type 450 (Polskie Młyny SA, Poland), sugar (Kupiec SA, Poland) and fresh eggs (Farmio SA, Poland) were purchased at local food market in Warsaw, solid vegetable shortening—Akobake MP was donated by AAK (Karshamns, Szwecja). The fat content was reduced by 5%, 10%, and 15% compared to that of the control sample without the fat replacement (0). The fat content was supplemented by soluble oat β-glucan (samples: A1, A2, A3) or yeast β-glucan—insoluble (samples: B1, B2, B3). The following types of β-glucan preparations were used: yeast (Biothera, Saint Paul, MN, USA) containing 91.31% of this biopolymer in dry mass (soluble fiber), and oat β-glucan (Y&L Biotech Co., Ltd., Xi’an, China) containing 71.00% of polysaccharide in dry mass (soluble fiber). 

### 3.2. Preparation of Shortbread Biscuits

Fat, sugar, and eggs were blended for 4 min using the Braun Multiquick kitchen processor (type 4644) to obtain a creamy texture. Then, flour and β-glucan were added. Thereafter, the mixed dough was rolled out and cut in square pieces 55 mm in length and 4 mm in thickness. The shortbread biscuits cookies were placed into baking cups and baked for 8 min. at 170 °C in UNOX type XBC convection oven (Vie Dell Ariginato, Padowa, Italy). Six batches of biscuits with β-glucan and sample of control were made and analyzed 24 h after baking. Each baking was repeated three times.

### 3.3. Analysis

#### 3.3.1. Analysis of Raw Fat

Gas chromatography was used to determine fatty acids (FA) composition of the investigated fats according to the ISO Standard [50]. Methyl esters were prepared according to the Polish standard [51]. Instrument: HP 6890 gas chromatography system with autosampler; SGE Capillary BPX 70, column 60 m × 0.25 mm ID; oven: temperature program from 160 to 210 °C, rate: 2.5 °C/min; air: 300 mL/min; carrier gas: helium, injector: Split-Splitless 240 °C; detector: FID 250 °C; software: HP Chemstation v. 3.11. Peaks of fatty acids and trans-isomers of fatty acids were identified by comparing them with the retention time of samples of fatty acid methyl esters (Supelco 37, Sigma Aldrich, St. Louis, MO, USA).

Atherosclerotic index (IA) indicates a lipid health quality, it was determined as the ratio of the sum of the major SFA to the sum of the major unsaturated FA [52]:(1)$$\mathrm{IA}=\frac{\left(4\cdot C14:0+\;C16:0+\;C18:0\right)}{\left(\Sigma \mathrm{MUFA}\;+\;\Sigma \mathrm{PUFA}\text{-}\omega \text{-}6+\;\Sigma \mathrm{PUFA}\text{-}\omega \text{-}3\right)}$$
where MUFA—monounsaturated fatty acids, and PUFA—polyunsaturated fatty acids.

Acid value and peroxide value were determined according to ISO standards [53,54], respectively. 

#### 3.3.2. Methods for Shortbread Biscuits Analysis

##### Biscuits Density

Biscuit density (Dn) was calculated based on the volume determined in rapeseed and weight of products. All measurements were carried out on the day after baking. The parameter was calculated from the formula [26,55]: (2)$$\mathrm{Dn}=\frac{M}{V}$$
where Dn—density of the biscuits (g/cm^3^); M—mass of the biscuits (g) and V—volume of the biscuits (cm^3^).

The final results of biscuit density were presented as a mean value of 18 determinations ± SD.

##### Geometry of Biscuits and Spread Ratio

Dimensions of products were determined: diameter (D) and height (H). Biscuit height was measured using an electronic caliper (TCM, type:234,990, Tchibo, Germany), in the highest point of the shortbread biscuits, after cooling at room temperature. Each variant was prepared twice, on different days, and six biscuits from each batch were measured (12 determinations) at the maximum point [26,56]. The spread ratio was calculated from the formula: (3)$$\mathrm{SI}=\frac{D}{H}$$
where SI—spread ratio of the biscuits (mm); diameter (D) (mm); and H—height of the biscuits (mm).

The final results were presented as a mean value of 12 determinations ± SD.

##### Color Assessment

The color intensity of shortbread biscuits surface with and without the addition of different concentrations of β-glucan was determined after baking using a tristimulus reflectance colorimeter (type:CR-200, Osaka, Japan). L*, a*, b* parameters were determined. Test was performed at three points on the product surface. The final result was the arithmetic mean of 12 measurements. 

CIEL*a*b* system was used, where: L* (lightness)—ranging from 0 (black) to 100 (white); a*—represents red (positive value) or green (negative value); and b* represents yellow (positive value) or blue of color (negative value). B* and a* ranging from −60 to +60.

The following parameter were calculated based on the results (ΔE—total color difference; BI—browning index; chroma—color saturation) [39,40]:(4)$$\Delta E=\sqrt{{\left(a_{c}^{*}-{\;a}_{x}^{*}\right)}^{2}+{\left(b_{c}^{*}-{\;b}_{x}^{*}\right)}^{2}+{\left(L_{c}^{*}-L_{x}^{*}\right)}^{2}}$$
where L*_c_, a*_c_, b*_c_ are the color parameters of the control sample and L*_x_, a*_x_, b*_x_ are the color parameters of the samples containing β-glucan.

Browning index (BI) was calculated as: (5)$$\mathrm{BI}=\frac{\left[100\cdot \left(x\;-0.31\right)\right]}{0.172}$$
where:(6)$$x=\frac{\left(a^{*}+1.75\cdot L^{*}\right)}{\left(5.645\cdot L^{*}+{\;a}^{*}-3.012\cdot b^{*}\right)}$$

Chrome was calculated as: (7)$$C\mathrm{hroma}\;=\sqrt{a^{*2}+b^{*2}}$$
where a*, b* are the color parameters of the study sample.

Moreover, C*after storage (modulus of difference between C*_1 day_ and C*_22 days_) was calculated:C*_after storage_ = ǀC*_1 day_ − C*_22 days_ǀ(8)
where C*_after storage_, C*_1 day_—chroma determined on biscuits after 1 day storage, C*_22 days_—Chroma determined on stored biscuits (22 days).

ΔE* after storage and BI* after storage was calculated analogically to the Equation (7).

##### Texture Analysis of Shortbread Biscuits

Texture of the products was analyzed by the instrumental method using a ZWICK 1120 texturemeter (Germany) using the Warner–Bratzler straight blade. Texture of shortbread biscuits was determined by three-point breaking test [26]. Procedure: broke force of cookies was determined as an indication of hardness. A 5 kg load cell was used and crosshead speed set at 3 mm/min, temperature 23 ± 2 °C. The assay was performed in nine replications.

##### Moisture Content

After 1, 8, 15, and 22 days, the samples were grinded in a mill and transferred (3 g) to a weighted moisture dishes and were placed in laboratory dryer at 103 ± 2 °C for 3 h [20] (Dryer SKP-100, Poland). Water content (%) was calculated as the difference in weight before and after drying. The analysis was determined in triplicate for each variant.

##### Sensory Evaluation of Shortbread Biscuits

Quantitative descriptive analysis (QDA) was used to carry out the detailed sensory assessment of the samples [39]. The sensory analysis was conducted by a trained 10-person team (24–48 years of age), made each evaluation in duplicate. The following QDA attributes were selected: appearance (color uniformity, and cracking), texture (hardness, crispness), odor (typical, oily, floury, and others), and taste (typical, oily, sweet, bitter, and others); moreover, overall quality was evaluated. The intensity of sensory attributes was evaluated on 100 mm undivided linear scales. The scales were anchored at each extreme with “not at all ….” and “extremely ….”. Samples were randomly selected from each batch and given in random order, each in a separate plastic box marked with a code.

### 3.4. Storage Test

The shortbread biscuits were stored for 8, 15, and 22 days at room temperature (21 ± 2 °C), in polyethylene bags. Color parameters, texture, and moisture content were analyzed. The analysis was determined in triplicate for each variant.

### 3.5. Statistical Analysis

The effects of the content of β-glucan replacement on properties of dough and shortbread biscuits were performed using one-way ANOVA and regression. The assessment of the significance of differences between the means was performed using Tukey’s test (*p* < 0.05). The regression method was used for model determination (x—the quantity of addition of β-glucan preparation). Statistical analysis was carried out with Statgraphics plus 4.0 package (Statistical Graphics Corp., Warreton, VA, USA).

## 4. Conclusions

The study showed that the replacement of shortening, which is a source of saturated fatty acids, by β-glucan in shortbread biscuits is possible to a limited extent. The decrease in energy value of the products (from 11.7 kcal to 36 kcal/100 g of the product) and the content of undesirable fatty acids (OFA) (maximum 2.1 g/100 g of the product) and the increase in the share of β-glucan, regardless of its origin (oat, yeast), worsened quality shortbread biscuits. Moreover, applied additives influenced dynamics of changes in physical parameters during storage of shortbread biscuits. Microbiological β-glucan had a more negative effect on quality of the products. The study showed that oat β-glucan can partially replace (5%) shortening in shortbread biscuits without adversely affecting their physical and sensory quality. While this may not be enough β-glucan to make a health claim (FDA and EFSA approved) on a product label, adding β-glucan to food instead of fat is a good way to improve your health by increasing dietary fiber, reducing calories, and saturated fatty acids (SFA) in the diet.

## Figures and Tables

**Figure 1 molecules-27-01393-f001:**
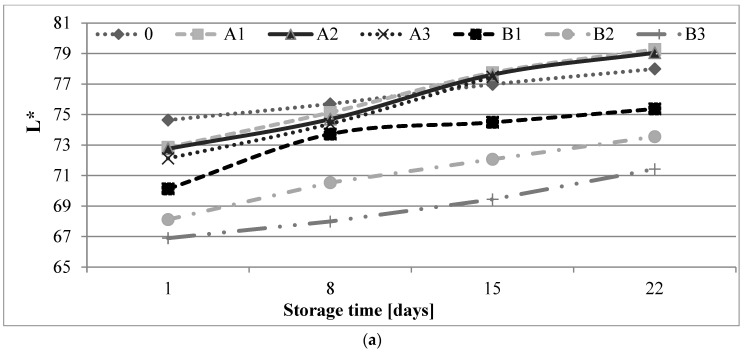
Changes in color parameters of shortbread biscuits during storage: (**a**) L*; (**b**) a*; and (**c**) b* variants descriptions are given under Table 2.

**Figure 2 molecules-27-01393-f002:**
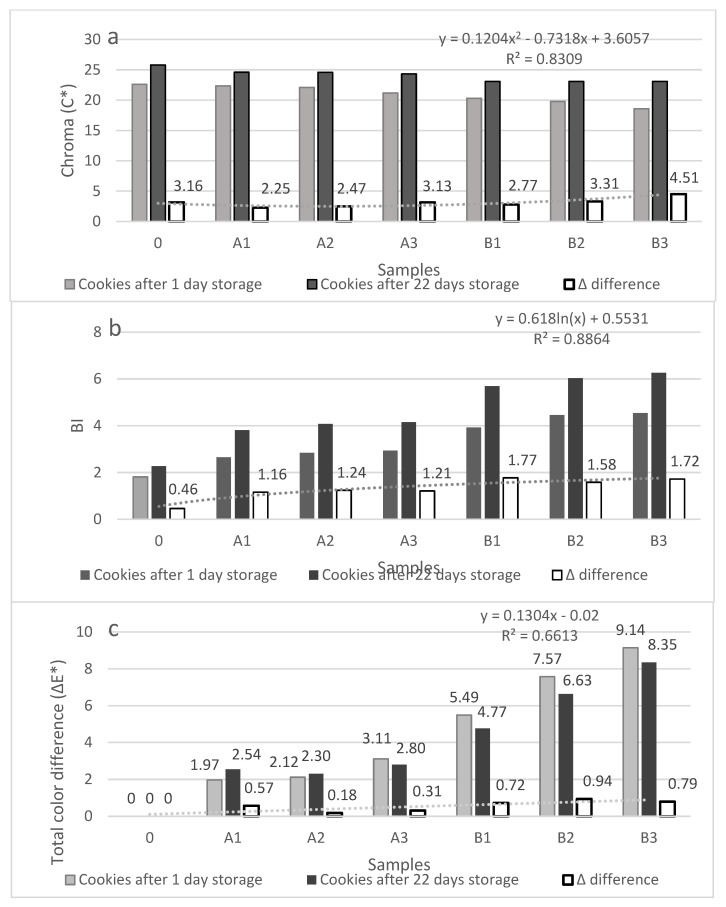
Shortbread biscuits color assessment: (**a**) chroma (C*); (**b**) browning index; and (**c**) total color difference (ΔE*).

**Figure 3 molecules-27-01393-f003:**
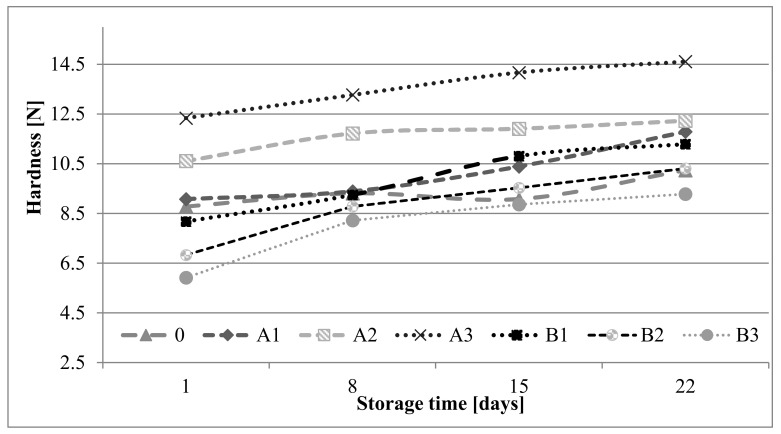
Changes of hardness in shortbread biscuits during storage.

**Figure 4 molecules-27-01393-f004:**
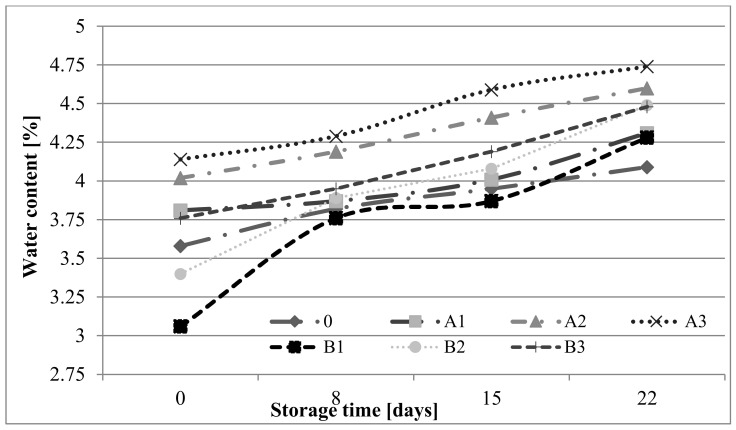
Water content changes in shortbread biscuits during storage.

**Figure 5 molecules-27-01393-f005:**
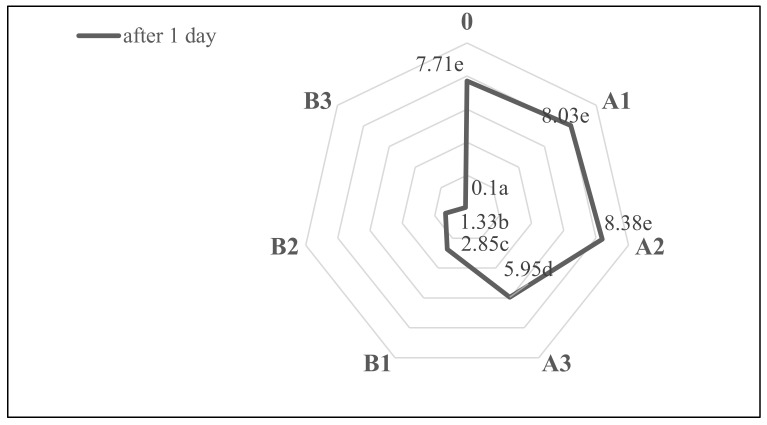
Effect of fat substitution with β-glucan on overall sensory quality of shortbread biscuits after baking. Different letters indicate significantly different values (*p* < 0.05).

**Table 1 molecules-27-01393-t001:** Characteristics of shortening and the atherosclerotic index (IA).

AV (mg KOH/g of Fat)	PV (meqO/kg of Fat)	Main Groups of Fatty Acids (FA) [%]				OFA	DFA	IA
		SFA	TFA	MUFA	Essential FA			
0.19 ± 0.01	0.74 ± 0.02	51.60 ± 0.21 (4.82%—stearic acid)	0.52 ± 0.05	39.94 ± 0.73 (39.02%—oleic acid)	7.94 ± 0.24	52.12	47.88	1.13

AV—acid value, PV—peroxide value, FA—fatty acids; SFA—saturated FA; TFA—trans FA; MUFA—monoenic FA; DFA—desirable FA; OFA—undesirable FA.

**Table 2 molecules-27-01393-t002:** Density and geometry of shortbread biscuits.

Samples	Fat Reduction (%)	Density (g/cm^3^)	Diameter (mm)	Hight (mm)	SI
0 (control)	0	0.72 a ± 0.02	56.18 d ± 0.87	6.14 cd ± 0.28	9.15
A1	5	0.72 a ± 0.01	55.96 d ± 0.58	6.30 de ± 0.20	8.88
A2	10	0.78 c ± 0.02	55.33 c ± 0.17	6.41 ef ± 0.23	8.63
A3	15	0.82 d ± 0.00	54.85 b ± 0.47	6.52 f ± 0.32	8.41
B1	5	0.75 b ± 0.02	55.09 bc ± 0.44	6.02 c ± 0.27	9.15
B2	10	0.77 c ± 0.01	54.31 a ± 0.24	5.80 b ± 0.26	9.36
B3	15	0.94 d ± 0.02	54.11 a ± 0.38	5.61 a ± 0.25	9.65

Different letters indicate significantly different values (*p* < 0.05). a, b, c, d—homogenous groups obtained by Tukey’s test (*p* < 0.05).

**Table 3 molecules-27-01393-t003:** Color parameter of shortbread biscuits after baking.

Sample	L*	a*	b*	ΔE	IB	Chrome
0	74.64 e ± 1.02	0.51 a ± 0.24	22.61 d ± 1.31	-	1.81	22.62
A1	72.86 d ± 1.81	1.31 b ± 0.52	22.31 d ± 1.09	1.97	2.65	22.35
A2	72.76 d ± 1.87	1.50 b ± 0.53	22.06 d ± 1.01	2.12	2.84	22.11
A3	72.13 d ± 1.93	1.58 b ± 0.47	21.12 c ± 1.50	3.11	2.94	21.18
B1	70.13 c ± 1.33	2.46 c ± 1.10	20.16 b ± 0.93	5.49	3.92	20.31
B2	68.12 b ± 1.26	2.85 c ± 0.86	19.56 b ± 0.97	7.57	4.45	19.77
B3	66.90 a ± 1.42	2.86 c ± 0.95	18.36 a ± 0.74	9.14	4.54	18.58

Different letters indicate significantly different values (*p* < 0.05). a, b, c, d, e—homogenous groups obtained by Tukey’s test (*p* < 0.05).

**Table 4 molecules-27-01393-t004:** Results of sensory analysis.

	Samples	0	A1	A2	A3	B1	B2	B3
Parameter		Appearance						
color uniformity		6.26 ± 1.41	6.33 ± 1.42	6.58 ± 1.71	6.74 ± 1.07	6.95 ± 1.33	7.45 ± 1.33	8.38 ± 1.14
cracking		0.88 ± 0.45	0.81 ± 0.56	2.08 ± 0.98	3.91 ± 1.02	2.91 ± 0.88	4.45 ± 1.23	4.75 ± 1.34
		Texture						
hardness		6.35 ± 1.51	5.64 ± 1.04	5.31 ± 1.24	3.13 ± 1.03	3.15 ± 1.00	2.05 ± 1.02	1.58 ± 0.67
crispness		8.34 ± 1.16	8.48 ± 1.01	8.39 ± 1.17	6.88 ± 1.03	7.10 ± 1.08	6.89 ± 0.98	4.9 ± 0.92
		Odor						
typical		7.58 ± 1.46	7.95 ± 1.27	8.1 ± 0.98	7.91 ± 2.06	5.93 ± 0.98	5.2 ± 1.02	2.44 ± 0.65
oily		6.7 ± 0.97	4.79 ± 1.02	3.03 ± 0.65	2.25 ± 1.21	5.05 ± 1.23	5.91 ± 1.72	7.25 ± 2.17
floury		1.69 ± 0.67	0.83 ± 0.24	1.51 ± 0.45	2.3 ± 1.02	1.08 ± 0.21	0.96 ± 0.22	0.75 ± 0.21
others		0 ± 0.00	0 ± 0.00	0.46 ± 0.23	0.08 ± 0.02	1.66 ± 0.54	4.8 ± 0.76	6.58 ± 1.54
		Taste						
typical		7.28 ± 1.21	7.86 ± 1.09	8.2 ± 0.82	6.76 ± 0.88	3.15 ± 1.02	2.18 ± 0.88	0.19 ± 0.08
oily		5.89 ± 0.72	4.64 ± 1.48	3.79 ± 1.05	2.19 ± 1.00	4.2 ± 1.13	2.26 ± 0.99	1.85 ± 1.01
sweet		5.94 ± 1.27	5.91 ± 0.77	5.01 ± 1.06	4.89 ± 1.18	3.48 ± 1.04	1.46 ± 1.05	0.85 ± 0.22
bitter		0.00 ± 0.00	0.00 ± 0.00	0.00 ± 0.00	0.00 ± 0.00	4.08 ± 1.02	7.28 ± 1.22	8.64 ± 2.89
others		0.00 ± 0.00	0.06 ± 0.02	0.56 ± 0.32	0.88 ± 0.12	4.74 ± 1.01	7.61 ± 1.32	9.4 ± 0.69

**Table 5 molecules-27-01393-t005:** Correlation analysis between β-glucan content and physical and sensory parameters of biscuits.

Parameter	Oat β-Glucan		Yeast β-Glucan	
	R	*p*-Value	r	*p*-Value
Weight	0.37	0.006	0.26	0.021
Width	−0.78	0	−0.88	0
Thickness	0.48	0	−0.62	0
Volume	−0.89	0	−0.91	0
Texture	−0.30	0.026	-	>0.5
Water content	0.96	0	0.35	0.091
	Color			
L*	-	>0.05	−0.88	0
a*	-	>0.05	-	>0.05
b*	−0.39	0.003	−0.82	0
	Sensory analysis			
Overall quality	−0.39	0.0274	−0.89	0

**Table 6 molecules-27-01393-t006:** Lipid content, fatty acids (FA) profile, and calorific value of shortbread biscuits (g/100 g of products).

Shortbread Biscuits	0	A1/B1	A2/B2	A3/B3
Fat (%)	27.0%	25.7	24.3	23.0
SFA (%)	13.9	13.2	12.5	11.8
MUFA+PUFA (%)	12.9	12.3	11.6	11.0
Caloric value decrease (kcal)	-	11.7	24.3	36.0

0—control sample (0% of β-glucan content and 100% of fat content); A1—sample with 1.3% of oat β-glucan content and with 5% reduction in fat content; A2—sample with 2.7% of oat β-glucan content and with 10% reduction in fat content; A3—sample with 4.0% of oat β-glucan content and with 15% reduction in fat content; B1—sample with 1.3% of yeast β-glucan content and with 5% reduction in fat content; B2—sample with 2.7% of yeast β-glucan content and with 10% reduction in fat content; B3—sample with 4.0% of yeast β-glucan content and with 15% reduction in fat content.

## Data Availability

The data presented in this study are available on request from the corresponding author.

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
