# Peer review of "Study on the Incorporation of Oat and Yeast β-Glucan into Shortbread Biscuits as a Basis for Designing Healthier and High Quality Food Products"

_molecules, 2022, doi:10.3390/molecules27041393_

Round 1

Reviewer 1 Report

Authors present "Study on the incorporation of oat and yeast β-glucan into short-bread biscuits as a basis for designing healthier and with good quality food products", however the "healthier" word in the title were not clear enough represented on the manuscript. Authors can complete the manuscirpt with data of remaining β-glucan on the biscuit that can make sure that β-glucan not only gave the appearance effects but also health or functional effects.

The manuscript only describe the role of β-glucan towards the short bread biscuits characteristics including the Physico-chemical and sensory characteristic. In some section mentioned that β-glucan subtituted fat, however in the research design and methodology the substitution was not reflected. The manuscript only compared the Oat and Yeast β-glucan regarding the characteristics of short bread biscuit.

Be consistent and focus with the treatment and parameter identified, so that reflected on title and background. 

Please check and re-draw the Graphic or Table so that it can be understood easily. The presented graphic and table was to crowded and confusing, it is not proper for publication. 

Please be clear in describing the methodology, cite some reference for the methodology to make sure that the method is valid.

Authors used the ANOVA, please explain how many treatment and replication that has been done? 

The explanation regarding the effect of β-glucan addition towards the parameter (weight, width, etc.) need to be highlighted and clearer on the background 

Author Response

Thank you very much for the constructive comments and suggestion. In my opinion after the introduction of all changes into the manuscript, our work became more readable with higher knowledge value.

According to Reviewers suggestions all corrections and information were placed in the manuscript. The modifications of the manuscript were made using the "track changes" option and red line. We have referred in detail to the comments of the Reviewers, which are marked below with an Italic. The answers to all comments are below.

Suggestion: Authors present "Study on the incorporation of oat and yeast β-glucan into short-bread biscuits as a basis for designing healthier and with good quality food products", however the "healthier" word in the title were not clear enough represented on the manuscript. Authors can complete the manuscirpt with data of remaining β-glucan on the biscuit that can make sure that β-glucan not only gave the appearance effects but also health or functional effects.”

Answer:

The discription of healthy value of oat and yeast β-glucan was described in the introduction, as well as positive effect of lowering fat content on energy value of biscuits.

The effect of β-glucan as a fat replacer on energy value was presented in Table 6.

The discription of healthy value of oat and yeast β-glucan was described in the introduction, as well as positive effect of lowering fat content on energy value of biscuits.

Suggestion: The manuscript only describe the role of β-glucan towards the short bread biscuits characteristics including the Physico-chemical and sensory characteristic. In some section mentioned that β-glucan subtituted fat, however in the research design and methodology the substitution was not reflected.

Answer:

The level of substitution of fat by β-glucan is metioned in the Materials and methods section, table 6, and in 2.2. section. Extended content is highlighted in red in the manuscript.

Suggestion: “Please check and re-draw the Graphic or Table so that it can be understood easily. The presented graphic and table was to crowded and confusing, it is not proper for publication”

Answer:

 Table 3 and Figure 4 were changed to make it easier to read and understand.

Suggestion:  „Please be clear in describing the methodology, cite some reference for the methodology to make sure that the method is valid.”

Answer:

In methodology section references were given, there where it was not cited. All changes were highlighted

Suggestion: “Authors used the ANOVA, please explain how many treatment and replication that has been done?”

Answer:

Regarding “Biscuits density, Geometry of biscuits, Color Assessment” - The final results were presented as a mean value of 12 determinations. Regarding “Texture analysis of shortbread biscuits” - The assay was performed in 9 replications. Regarding the water content in the storage test -The analysis was determined in triplicate in each variant

Suggestion: “The explanation regarding the effect of β-glucan addition towards the parameter (weight, width, etc.) need to be highlighted and clearer on the background”

Answer:

In subsection 3.1. Health value of lipids in short-dough biscuits and their density and geometry ”more attention was devoted to explaining the influence of β-glucan addition on the discussed physical parameters of cookies. Extended content is highlighted in red in the manuscript.

Reviewer 2 Report

Dear Authors:

The work presented in the manuscript is interesting, however, in its current form there is a need to work on the following issues

  1. English language, a thorough revision of the language is required e.g in the abstract this sentence "limited in fat as the most caloric component of food and often a source of unsafe saturated fatty 20 acids (FA) and trans isomers" is incomplete. Similar corrections are to be made to the whole manuscript.         
  2. Line 108: "β-glucan is poorly utilized in human digestion tract and, therefore, functions as a non-caloric food" the word poorly shows disadvantages rather than benefits, so I would suggest reframing the sentence. 
  3. line 129-130 "The most similar to the control sample were products in which fat was reduced 129 by 5% and in its place oat β-glucan was introduced" reframe the sentence. ......
  4. line 13-131 "Worse effects were achieved with the 130 use of microbial β-glucan" reframe the sentence. 
  5. In table 3, I suggest not to write SD, instead, write +/- after the mean value
  6. Figure 4, please reframe the presentation of the data, make it easily understandable for the readers
  7. The discussion part needs to be improved, authors have described results properly with the support of the references but the reasoning part including scientific merit is lacking. 

Author Response

Thank you very much for the constructive comments and suggestion. In my opinion after the introduction of all changes into the manuscript, our work became more readable with higher knowledge value.

According to Reviewers suggestions all corrections and information were placed in the manuscript. The modifications of the manuscript were made using the "track changes" option and red line. We have referred in detail to the comments of the Reviewers, which are marked below with an Italic. The answers to all comments are below.

Suggestion 1: “English language, a thorough revision of the language is required e.g in the abstract this sentence "limited in fat as the most caloric component of food and often a source of unsafe saturated fatty acids (FA) and trans isomers" is incomplete. Similar corrections are to be made to the whole manuscript.”

Answer:

The work has been corrected in terms of linguistic correctness. Changes have been made to abstracts and the rest of the manuscript.

Suggestion 2:”Line 108: "β-glucan is poorly utilized in human digestion tract and, therefore, functions as a non-caloric food" the word poorly shows disadvantages rather than benefits, so I would suggest reframing the sentence.” 

Answer:

The sentence was reframed

Suggestion 3: “line 129-130 "The most similar to the control sample were products in which fat was reduced by 5% and in its place oat β-glucan was introduced" reframe the sentence.”

Answer:

The sentence was reframed

Suggestion 4: “line 13-131 "Worse effects were achieved with the use of microbial β-glucan" reframe the sentence.” 

Answer:

The sentence was reframed

Suggestion 5: “In table 3, I suggest not to write SD, instead, write +/- after the mean value”

Answer:

Table 3 was corrected as suggested by the reviewer

Suggestion 6: “Figure 4, please reframe the presentation of the data, make it easily understandable for the readers”

Answer:

Figure 4 has been changed to improve its readability

Suggestion 7: “The discussion part needs to be improved, authors have described results properly with the support of the references but the reasoning part including scientific merit is lacking.”

Answer:

The discussion is improved. Added content is highlighted in red.

Reviewer 3 Report

It’s a nice try to replace shortening by beta-glucans from oat and yeast. The findings are interesting and offer good perspectives as to reduce the fat content in the daily diet. However, the substitution caused deterioration of biscuits quality and affected changes during storage.

Overall, experiments were properly designed and conducted, leading to conclusive results. However, the manuscript is not clearly written. I have a few specific comments listed hereafter, which I hope will help improving even further the quality of the manuscript.

In the Introduction, more background information should be provided, such as why choose beta-glucan to reduce the fat in short-bread biscuits, why choose the two sources of beta-glucan, and introduce the related research progress.

Line 54: Trans Fatty Acid/TFA may be not only responsible for overweight and obesity. The appropriate references should be added here. The same goes for other places like line 61-64. Please look through the full manuscript.

Table 1: There is no PUFA in the table. The abbreviation for PUFA could be deleted.

Line 104/140/203/230/245/268: Please change “3.1-3.7” to “2.2-2.7”.

Table 2 was not results and should be put in the M&M.

Line 130: “worse effects were achieved with the use of microbial beta-glucan.” Could you discuss the possible reason, please?

Author Response

Thank you very much for the constructive comments and suggestion. In my opinion after the introduction of all changes into the manuscript, our work became more readable with higher knowledge value.

According to Reviewers suggestions all corrections and information were placed in the manuscript. The modifications of the manuscript were made using the "track changes" option and red line. We have referred in detail to the comments of the Reviewers, which are marked below with an Italic. The answers to all comments are below.

Suggestion: In the Introduction, more background information should be provided, such as why choose beta-glucan to reduce the fat in short-bread biscuits, why choose the two sources of beta-glucan, and introduce the related research progress.

Answer:

The introduction is improved. Added content is highlighted in red.

 Suggestion: „Line 54: Trans Fatty Acid/TFA may be not only responsible for overweight and obesity. The appropriate references should be added here. The same goes for other places like line 61-64. Please look through the full manuscript.”

Appropriate references were added.

Brayner B., Kaur G., Keske M.A, Perez-Cornago A., Piernas C., Livingstone K.M, Dietary Patterns Characterized by Fat Type in Association with Obesity and Type 2 Diabetes: A Longitudinal Study of UK Biobank Participants, The Journal of Nutrition, V.151, Issue 11, November 2021, Pages 3570–3578,

Onacik-Gur, S.; Żbikowska, A.; Kapler, E.; Kowalska H. Effect of barley glucan addition as a fat replacer on muffin quality. Acta. Sci. Pol. Technol. Aliment. 2016, 15(3), 247-256.

Vetvicka, V.; Vetvickova, J. Physiological effects of different types of beta-glucan. Biomed. Pap. Med. Fac. Palacky. Univ. Olomouc 2007, 151, 225–231.

 Suggestion: Table 1: There is no PUFA in the table. The abbreviation for PUFA could be deleted.

The abbreviation for PUFA is deleted

Suggestion: Line 104/140/203/230/245/268: Please change “3.1-3.7” to “2.2-2.7”.

Answer:

Subsections numbers were changed

Suggestion: „Table 2 was not results and should be put in the M&M.

Table 2 is moved to M&M section.

Suggestion: „ Line 130: “worse effects were achieved with the use of microbial beta-glucan.” Could you discuss the possible reason, please?”

The authors improved the discussion. Added content is highlighted in red.

„The reason of higher density of products with microbial than oat β-glucan is its low ability to dilute in water. Particles of microbial β-glucan weigh biscuits down and did not let gas cells to expand during baking.”

Round 2

Reviewer 3 Report

I have no further questions.